# Diagnostic Utility of a Combined MPO/D-Dimer Score to Distinguish Abdominal Aortic Aneurysm from Peripheral Artery Disease

**DOI:** 10.3390/jcm12247558

**Published:** 2023-12-07

**Authors:** Branislav Zagrapan, Johannes Klopf, Nihan Dide Celem, Annika Brandau, Patrick Rossi, Yulia Gordeeva, Alexandra Regina Szewczyk, Linda Liu, Diana Ahmadi-Fazel, Sina Najarnia, Lukas Fuchs, Hubert Hayden, Christian Loewe, Wolf Eilenberg, Christoph Neumayer, Christine Brostjan

**Affiliations:** 1Department of General Surgery, Division of Vascular Surgery, University Hospital Vienna, Medical University of Vienna, 1090 Vienna, Austria; branislav.zagrapan@unibe.ch (B.Z.); johannes.klopf@meduniwien.ac.at (J.K.); celem.nihan.dide@gmail.com (N.D.C.); annikabrandau@gmx.de (A.B.); n1542485@students.meduniwien.ac.at (P.R.); yuli.grdv@gmail.com (Y.G.); szew.alex@gmail.com (A.R.S.); n11774105@students.meduniwien.ac.at (L.L.); n01549411@students.meduniwien.ac.at (D.A.-F.); sina.najarnia@gmail.com (S.N.); n1248453@students.meduniwien.ac.at (L.F.); hubert.hayden@meduniwien.ac.at (H.H.); wolf.eilenberg@meduniwien.ac.at (W.E.); christoph.neumayer@meduniwien.ac.at (C.N.); 2Department for Bioimaging and Image-Guided Therapy, Division of Cardiovascular and Interventional Radiology, University Hospital Vienna, Medical University of Vienna, 1090 Vienna, Austria; christian.loewe@meduniwien.ac.at

**Keywords:** abdominal aortic aneurysm, peripheral artery disease, D-dimer, myeloperoxidase, MPO/D-dimer score, diagnosis, biomarker

## Abstract

Abdominal aortic aneurysm (AAA) and peripheral artery disease (PAD) share pathophysiological mechanisms including the activation of the fibrinolytic and innate immune system, which explains the analysis of D-dimer and myeloperoxidase (MPO) in both conditions. This study evaluates the diagnostic marker potential of both variables separately and as a combined MPO/D-dimer score for identifying patients with AAA versus healthy individuals or patients with PAD. Plasma levels of MPO and D-dimer were increased in PAD and AAA compared to healthy controls (median for MPO: 13.63 ng/mL [AAA] vs. 11.74 ng/mL [PAD] vs. 9.16 ng/mL [healthy], D-dimer: 1.27 μg/mL [AAA] vs. 0.58 μg/mL [PAD] vs. 0.38 μg/mL [healthy]). The combined MPO/D-dimer score (median 1.26 [AAA] vs. −0.19 [PAD] vs. −0.93 [healthy]) showed an improved performance in distinguishing AAA from PAD when analysed using the receiver operating characteristic curve (area under the curve) for AAA against the pooled data of healthy controls + PAD: 0.728 [MPO], 0.749 [D-dimer], 0.801 [score]. Diagnostic sensitivity and specificity ranged at 82.9% and 70.2% (for score cut-off = 0). These findings were confirmed for a separate collective of AAA patients with 35% simultaneous PAD. Thus, evaluating MPO together with D-dimer in a simple score may be useful for diagnostic detection and the distinction of AAA from athero-occlusive diseases like PAD.

## 1. Introduction

The diagnosis of abdominal aortic aneurysm (AAA) is based on imaging and is often incidental (upper abdominal sonography or computed tomography for reasons initially unrelated to AAA) since it frequently occurs without major symptoms [1]. The benefits of a systematic population screening for AAA have been the subject of discussions in the published literature. So far, it has been adopted in only a few countries (such as Sweden, Germany, UK and the USA) whereby the guideline-recommended screening may also be underutilised [2]. Additionally, there seem to exist relevant differences in the potential benefits of such screening between men and women [3]. To maximise the health benefits of an AAA screening programme, the target population should be well selected [4] and, from what is known from epidemiological studies, likely would comprise men aged 65–75 with a history of tobacco use [3,5,6]. In addition to the central role of imaging in AAA diagnosis and monitoring, several potential molecular biomarkers have been characterised which can be measured in peripheral blood and used for identifying patients at high risk of AAA development or progression [7,8]. Several studies have shown a strong association of AAA with markers of hemostatic system activation, in particular D-dimer [9,10]. This test is widely available and used primarily in the context of deep vein thrombosis and/or venous thromboembolism. The interpretation is not always straightforward as the cut-off for an abnormal D-dimer level varies with age and the specificity may be confounded by many conditions in which the hemostatic and fibrinolytic systems are activated [11]. Furthermore, high circulating D-dimer has been significantly associated with various atherosclerosis-related conditions in addition to AAA, including peripheral artery disease (PAD) [12], coronary artery disease and ischemic stroke [13,14]. In other words, in order to be useful, increased D-dimer needs to be interpreted within the clinical context.

In contrast to D-dimer, peripheral blood myeloperoxidase (MPO) is not typically assessed in routine clinical practice, but nevertheless it is a generally and easily available parameter. This leukocyte-derived (mainly neutrophil and monocyte/macrophage) enzyme with a primary bactericidal function is also implicated in the process of atherogenesis and has been studied as a potential biomarker of atherosclerotic risk/extent mainly in the context of coronary artery disease [15,16,17]. In the context of peripheral artery disease, high MPO has been associated with poorer outcomes and suggested as a biomarker for risk stratification [18,19]. The published research on MPO in the setting of AAA has also shown association with AAA presence and growth [20].

Given the limited specificity of D-dimer and MPO on their own, we previously proposed a combined MPO/D-dimer score which has shown improved performance (compared to the single parameters) in a diagnostic setting [21]. While D-dimer was more closely associated with the dimensions of AAA-related thrombi, the newly developed score showed the highest correlation with AAA diameter and with rapid AAA growth [21]. The present report evaluates the performance of the MPO/D-dimer score in the context of AAA vs. PAD, i.e., addresses the question of whether the score is suited to distinguishing abdominal aortic aneurysm from this athero-occlusive disease.

## 2. Methods

### 2.1. Study Design

The data analysed in this report originate from a prospective case–control observational study performed at the Vienna General Hospital between 2014 and 2021. Four cohorts of subjects were compared: patients with known AAA without known PAD (AAA1), patients with known AAA some of whom had PAD as a comorbidity (mixed cohort, AAA2), patients with PAD (but without aortic aneurysm) and a healthy (non-AAA, non-PAD) control group. All participants gave informed consent to be included in the study, which was approved by the Ethics Committee of the Medical University of Vienna (no. 1729/2014) and conducted according to the principles of the Declaration of Helsinki and its current amendments.

The inclusion criterion for the AAA cohorts 1 and 2 was an AAA diagnosis with a maximum abdominal aortic diameter of ≥3 cm (of non-mycotic and non-inflammatory subtype and unrelated to connective tissue disorders). PAD patients with a confirmed diagnosis in healthcare records and a pelvis/leg CTA scan were enrolled (20% stage I, 3% stage IIa, 60% stage IIb and 17% stage IV). The healthy control group was primarily recruited from general surgery, ophthalmology and urology patients (presenting for routine check-ups). As they were age-matched with the AAA collective, chronic conditions frequent in advanced age (such as hypertension, hyperlipidaemia or diabetes) were not considered an indication for exclusion. Yet, clinically manifest cardiovascular disease in terms of acute events and interventions (myocardial infarction, stroke, vascular stents, coronary artery bypass grafts) or clinically documented PAD and coronary heart disease were applied as exclusion criteria.

The aneurysm morphology of AAA patients was determined using computed tomography angiography (CTA) scans of the thoracic and abdominal region. To exclude the presence of AAA in the healthy controls and PAD patients, an ultrasound analysis was performed at the time of study inclusion, unless a recent (≤6 months) pelvis/leg CTA scan or comparable imaging record was available.

The exclusion criteria, applied to all groups except AAA2, were as follows: organ transplantation, recent cancer and/or chemotherapy in the past year, autoimmune disease and chronic haematological disease.

The demographics of all participants were collected via a structured questionnaire. Medical diagnosis and drug consumption were based on both a structured questionnaire for patient self-reporting and medical records; discrepancies were solved by contacting patients or primary care physicians. Distinct from our previous study [21], one-to-one matching was performed among the 4 groups based on age (±2 years) and sex.

### 2.2. Analysis of Blood Parameters

The D-dimer values were obtained as part of a routine set of diagnostic tests available from the Vienna General Hospital central laboratory service. Additionally, platelet-free plasma was prepared using a previously described protocol [22] within 60 min of peripheral blood collection into pre-chilled CTAD (citrate, theophylline, adenosine and dipyridamole) tubes and was stored in aliquots at −80 °C. The plasma MPO levels were measured using ELISA as per manufacturer’s instructions (R&D Systems, Minneapolis, MN, USA).

The combined MPO/D-dimer score was calculated using the formula “score” = −3.442 + 1.375*[D-dimer] + 0.205*[MPO], as previously described [21]. The diagnostic cut-off was set at zero, i.e., the score was originally designed to discriminate between AAA patients (score ≥ 0) and healthy controls (score < 0).

### 2.3. Statistical Analysis

The statistical analysis was performed in SPSS (Version 26, SPSS Inc., Chicago, IL, USA). A *p*-value < 0.05 was considered statistically significant. In general, metric data are presented as medians and interquartile ranges (IQRs). Categorical variables are given in count n and percent of total, and are assessed using chi square or Fisher’s exact tests. For statistical group comparisons, Wilcoxon’s rank-sum test was employed; multivariable analysis was performed using binary logistic regression. The diagnostic marker potential was assessed using the receiver operating characteristic curve (ROC) and C statistics.

## 3. Results

### 3.1. Patient Cohorts Show Comparable Demographics and Co-Morbidities While Healthy Controls Differ Significantly

In total, 41, 43, 41 and 37 persons were included in the four study groups healthy, PAD, AAA1 and AAA2, respectively. The proportion of female participants was 22.0%, 20.9%, 22.0% and 16.2%, respectively (Table 1). The number of current smokers was higher among the PAD, AAA1 and AAA2 groups as compared to the healthy control group, a difference in the smoking habit which was even more pronounced when expressed as the median (IQR) number of pack-years 37.0 (45.0), 38.0 (35.0) and 40 (18.0) vs. 8.5 (27.5). The three patient cohorts also had higher rates n (%) of hypertension: 39 (90.7%), 38 (92.7%) and 28 (75.7%) vs. 22 (53.7%) and hyperlipidaemia: 31 (72.1%), 34 (82.9%) and 32 (86.5%) vs. 10 (25%), respectively (PAD, AAA1 and AAA2 vs. healthy). Regarding routine blood parameters, PAD patients showed the lowest levels of blood cholesterol, while the controls presented with the highest median value, likely reflecting the use of lipid-lowering agents (17% versus 91% in the respective cohorts). Yet, the median values of blood lipids, HbA1c and C-reactive protein were within normal range for all four study groups.

The median maximum diameters of the aneurysm in the AAA groups were 51.8 mm (range 31.2–96.5 mm, AAA1) and 47.8 mm (range 32.2–62.8 mm, AAA2), respectively (Table 1). Both the AAA1 and AAA2 collectives included three patients with simultaneous AAA and a thoracic aortic aneurysm. The morphology of the abdominal aneurysms was predominantly fusiform (66% and 85%, AAA1 and AAA2, respectively) with infrarenal topography (76% and 79%, AAA1 and AAA2, respectively).

### 3.2. The Combined Score Is Superior to Single-Parameter Analysis in Discriminating AAA from PAD Patients

The plasma level of MPO was found to be significantly increased in both AAA cohorts compared to the healthy collective, as well as the PAD collective (median 13.63 ng/mL or 14.36 ng/mL, respectively (IQR = 11.27 or 9.65, respectively) vs. 9.16 ng/mL (IQR 7.06, healthy) or 11.74 ng/mL (IQR 4.93, PAD), respectively (Table 2, Figure 1A)).

Similarly, the median D-dimer level was increased more than twice in both AAA groups compared to the healthy group, ranging at 1.27 µg/mL (IQR 1.18) or 0.90 µg/mL (IQR 1.08), AAA1 and AAA2, respectively vs. 0.38 µg/mL (IQR 0.51), and was also higher than in PAD patients (0.58 µg/mL (IQR 0.55), Table 2, Figure 1B).

An increase in the concentration of both markers was found in the PAD vs. healthy group as well, albeit less pronounced than for the two AAA groups: MPO 11.74 ng/mL (IQR 4.93) vs. 9.16 ng/mL (IQR 7.06); D-dimer 0.58 µg/mL (IQR 0.55) vs. 0.38 µg/mL (IQR 0.51). In contrast to D-dimer, the difference in MPO between the healthy and PAD groups missed the cut-off for statistical significance (*p* = 0.055, Table 2). Of note, compared to the PAD patients, there was a statistically significant increase in MPO in both AAA groups, while for D-dimer, the cut-off for a statistically significant difference over PAD was only reached in AAA1 but not in the mixed AAA2 cohort.

Importantly, the difference in the combined MPO/D-dimer score was statistically significant when comparing the AAA groups and healthy controls (*p* < 0.001) as well as both AAA1 and AAA2 vs. PAD: median score 1.26 (IQR 3.95) and 1.07 (IQR 2.69) vs. −0.19 (IQR 1.44, PAD), *p* < 0.001 and *p* = 0.014, respectively (Table 2, Figure 1C).

### 3.3. The Score Prevails as an Independent Diagnostic Parameter for AAA

To further characterise the diagnostic marker value of the MPO/D-dimer score to discern AAA from both a healthy state (without cardiovascular burden) and athero-occlusive disease (PAD), multivariable analysis was conducted for the AAA1 group against the combined data of the healthy and PAD groups. The dichotomised score (i.e., score < 0 vs. score ≥ 0), together with smoking, hyperlipidaemia and coronary heart disease, prevailed as an independent risk factor associated with AAA diagnosis (Table 3).

Finally, the performance of the combined MPO/D-dimer score versus the two variables separately was assessed using ROC analysis. When compared to the combined “non-AAA” group, i.e., healthy + PAD, the score outperformed the separate variables for both AAA1 and AAA2: area under the ROC curve (AUROC) 0.801 (score) vs. 0.728 (MPO) or 0.749 (D-dimer) for AAA1 vs. non-AAA (Figure 2A); AUROC 0.736 (score) vs. 0.686 (MPO) or 0.709 (D-dimer) for AAA2 vs. non-AAA (Figure 2B). Based on the C statistics, the ROC curves differed significantly between the score and MPO (AAA1: *p* = 0.014, AAA2: *p* = 0.056). The previously set cut-off at score = 0 [21] resulted in a diagnostic specificity for AAA of 70.2% with a sensitivity of 82.9% (AAA1 vs. non-AAA = healthy + PAD) and a specificity of 70.2% and sensitivity of 67.6% when evaluating AAA2 vs. non-AAA = healthy + PAD patients.

## 4. Discussion

The association of an increased plasma D-dimer level with both PAD and AAA is well known [12,23]. The aim of the present report was to investigate the utility of a combined MPO/D-dimer score in a diagnostic setting where AAA and PAD exist either separately or as comorbidities. Hence, in addition to the stringently recruited AAA1 cohort which excluded patients with PAD, a second AAA group was compared in the analysis (AAA2) which included 13 patients (35.1%) with PAD as a comorbidity. The design of the two AAA collectives was based on the following rationale: cohort AAA1 did not include patients with diagnosed PAD to allow for a clearer separation between individuals with clinically manifest aneurysmal versus peripheral athero-occlusive disease in exploratory biomarker analysis. The exclusion criteria for cohort AAA2 were less stringent to reflect the usual frequency of concurrent AAA and PAD and to validate the diagnostic biomarkers under comorbidity conditions. Yet, with respect to the overall atherosclerotic burden, the AAA and PAD collectives did not differ in the frequency of coronary heart disease, myocardial infarction or stroke (Table 1).

Both AAA and PAD share certain pathophysiological features such as a chronic inflammatory state and the activation of the coagulation and fibrinolytic system, which provides the theoretical basis for D-dimer and MPO as potential disease markers [24,25]. Yet, the two disease conditions also differ in various aspects of their pathogenesis. While the blood vessel wall is largely devoid of the elastic recoil properties in both disorders, AAA is mainly driven by an expansile and degenerative process [26] whereas PAD is characterised by a stenotic process resulting from atherosclerotic occlusion of the blood vessel lumen [27]. Thus, the lipidomic profile has been suggested to differ between AAA and PAD [28]. With respect to vessel wall destruction, extracellular-matrix-degrading factors such as matrix metalloproteinases (MMP)-2 and 9 and their inhibitors may be more prominent in AAA than PAD. Yet, the potential of serum MMP levels as biomarkers seems limited for either condition [29,30].

The nature of the inflammatory infiltrate also differs between AAA and PAD: while it prominently features neutrophils and monocytes in AAA [31,32,33,34], monocytes and macrophages seem to be more relevant in atherosclerosis. Still, neutrophils and neutrophil-derived factors have a transient role in the atherosclerotic plaque “life cycle”, e.g., in plaque rupture [35,36]. Consequently, the profile of circulating inflammatory factors overlaps in part between AAA and PAD, i.e., C-reactive protein, interleukin (IL)-1 and IL-6 were consistently found associated with both conditions [37,38,39]. Yet, serum complement factor C5a, which functions as a chemotactic signal for neutrophils, is reportedly higher in AAA compared to PAD patients [40,41]. Also, various other neutrophil-derived mediators have been investigated as potential diagnostic biomarkers. However, direct comparisons between AAA and athero-occlusive disease are largely missing.

A key distinction between the two conditions is the often quite voluminous AAA-associated intraluminal thrombus (ILT), which is known to entrap leukocytes, thereby acting as a reservoir of many inflammatory mediators and extracellular-matrix-degrading enzymes [42]. With respect to the chronic setting and size of the ILT, higher circulating levels of D-dimer and MPO in AAA compared to PAD patients seem feasible. Yet, acute events such as atherosclerotic plaque rupture associated with subsequent ischemia must be recognised as potential confounding factors for the interpretation of D-dimer, MPO, as well as the combined score. A prominent example would be an acute coronary event leading to myocardial infarction, in which case the plasma concentration of MPO may be about 10 times as high as the values measured in our study [15]. Similarly, circulating D-dimer is also found increased in myocardial infarction, averaging at similar values as reported in our study [43,44]. Thus, any condition causing a major acute inflammatory response with neutrophil activation, or a thrombotic event associated with a spike in D-dimer, needs to be taken into consideration when interpreting the MPO/D-dimer based score in relation to PAD and AAA.

In the setting of chronic disease, our study showed a median increase in the circulating plasma levels of both D-dimer and MPO in all patient groups compared to the healthy controls, which is in line with previously published results and suggests an activation of the thrombogenic/fibrinolytic and innate immune systems with chronic degranulation of the myeloid cells in both conditions [12,18,19,20,23,45]. While plasma MPO was less suited to distinguishing PAD patients from healthy controls (*p* = 0.055), D-dimer failed to differentiate between the PAD and AAA2 cohorts. Importantly, the combined score was significantly higher in both AAA patient cohorts over the PAD group, as well as the healthy control cohort. Since the prevalence of cardiovascular comorbidities among the patient collectives was similar (Table 1) and the score prevailed in multivariable analysis (Table 3), this suggests a relevant increase in the measured variables, which may be leveraged in the chronic AAA setting. Of note, while the body mass index was slightly higher for AAA than PAD patients, it was not associated with the explored plasma parameters.

In the present study, the score showed an improved AUROC: 0.801 vs. 0.728 or 0.749 for MPO or D-dimer, respectively (AAA1 vs. “non-AAA”, i.e., healthy + PAD). Also, for the mixed collective of AAA and AAA with concomitant PAD (AAA2), the score resulted in a better AUROC compared to D-dimer or MPO alone (AUROC 0.736 vs. 0.709 or 0.686, respectively). In a recent report by Cai et al., D-dimer has similarly been proposed as a possible marker associated with AAA diagnosis in concomitant PAD (AUROC 0.703), with a maximum specificity of 85% and sensitivity of 77% [46]. A D-dimer level of >0.675 µg/mL was strongly associated with the presence of AAA in PAD patients, which is in line with our observation of a significant D-dimer increase in AAA (in our study, median 1.27 µg/mL (AAA1) vs. 0.90 µg/mL (AAA2) vs. 0.58 µg/mL (PAD) vs. 0.38 µg/mL (healthy), respectively). In a similar fashion, Memon et al. analysed potential biomarkers including MPO for AAA (compared against a non-AAA control group matched for comorbidities) with a resulting MPO AUROC of 0.71, specificity 59% and sensitivity 80% [20]. Our data suggest the improved performance of the combined MPO/D-dimer score in a diagnostic setting of AAA versus PAD (or healthy), showing an AUROC of 0.801 and specificity at 70% and sensitivity at 83% for the score cut-off set to zero.

Of note, the optimal one-to-one matching of cases, especially with respect to potential confounding variables, was a limitation in our study. Smoking is a major risk factor for atherosclerotic disease in general and is also strongly associated with AAA as well as PAD, which nearly precludes perfect matching with a healthy cohort with respect to smoking pack-years. Regarding the healthy controls and the PAD patients, only the absence of aortic aneurysms was verified. Peripheral arterial aneurysms were not considered an exclusion criterion for any of the three study cohorts. Unfortunately, segmental blood pressure and ankle–brachial index (to further document peripheral athero-occlusive disease) were not systematically recorded for the study groups.

In summary, (1) both D-dimer and MPO are increased in PAD and AAA, and (2) the circulating levels of these markers tend to be higher in AAA than PAD; thus, (3) combining the two markers in a calculated MPO/D-dimer score provides improved sensitivity and specificity with respect to AAA diagnosis, also in the setting of concomitant PAD. On this note, applying the MPO/D-dimer score with a cut-off value of 0 may prove helpful. In daily clinical practice, with increasing workloads and complex diagnostic situations, diagnostic scores may provide useful signals to alert clinicians to a potentially high-risk condition such as AAA and trigger specialist imaging. Our data suggest that it may be possible to use simple parameters to identify patients with AAA even among individuals with advanced vascular comorbidity such as PAD. While D-dimer is a routine parameter ordered frequently in patients with suspected or known cardiovascular conditions, plasma MPO is less commonly determined but widely available in clinical laboratories. The combined expenses of these diagnostic tests is within the cost range of abdominal ultrasound, and considerably lower than for computed tomography. However, the performance of these blood parameters and their combined score needs to be further evaluated in larger multicentre studies, against other circulating biomarkers and physical examination in a blinded manner, to confirm the usefulness of the proposed score as a simple pre-imaging test.

## Figures and Tables

**Figure 1 jcm-12-07558-f001:**
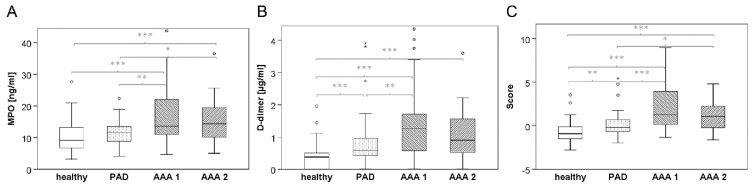
Boxplot illustration for plasma levels of (**A**) MPO, (**B**) D-dimer and (**C**) the combined score in the four investigated study cohorts. Of note, graphs of boxplots show the interquartile range for best resolution, while several outliers (black circles) or extreme values (black asterisks) are not depicted (i.e., not within the given scale). *p*-values for group comparisons (based on Wilcoxon’s rank-sum test) are given in Table 2 and illustrated in grey color: * *p* < 0.05, ** *p* < 0.01, *** *p* < 0.001.

**Figure 2 jcm-12-07558-f002:**
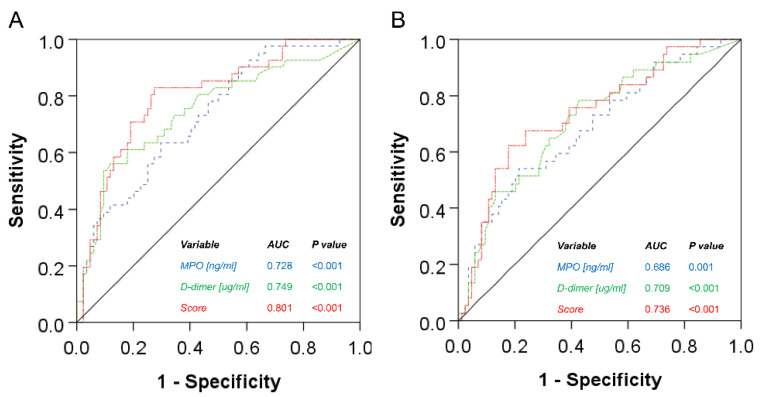
ROC analysis of the MPO/D-dimer score and the two variables separately for (**A**) AAA1 and (**B**) AAA2 groups against the combined group of healthy + PAD.

**Table 1 jcm-12-07558-t001:** Demographics and routine laboratory parameters of the study groups.

*Parameter*			*Healthy* *(n = 41)*	*PAD* *(n = 43)*	*AAA Cohort 1* *(n = 41)*	*AAA Cohort 2* *(n = 37)*	*p-Value*
*Metric variables*		n	Median (IQR)	Median (IQR)	Median (IQR)	Median (IQR)	
*Age [years]*		41/43/41/37	71.4 (13.3)	72.3 (11.9)	71.8 (12.4)	69.2 (11.7)	n.s. ^a^
*Smoking pack-years [py]*		40/43/39/35	8.5 (27.5)	37.0 (45.0)	38.0 (35.0)	40.0 (18.0)	H:P—<0.001 ^a^ H:A1—<0.001 ^a^ H:A2—<0.001 ^a^
*Body mass index*		32/43/40/37	26.1 (4.6)	25.8 (3.7)	27.6 (5.6)	26.8 (6.5)	P:A1—0.021 ^a^ P:A2—0.011 ^a^
*Thrombocytes [10^9^/l]*		40/42/38/34	239 (102)	228 (105)	208 (65)	218 (79)	H:A1—0.030 ^a^
*Leukocytes [10^9^/l]*		40/42/38/34	6.99 (3.21)	7.48 (2.93)	6.96 (2.43)	7.40 (3.37)	n.s. ^a^
*Neutrophils [10^9^/l]*		39/38/38/31	4.3 (2.4)	4.7 (2.0)	4.7 (2.1)	4.3 (2.6)	n.s. ^a^
*Fibrinogen [mg/dl]*		39/42/38/34	359 (90)	342 (90)	354 (141)	361 (90)	n.s. ^a^
*Cholesterol total [mg/dl]*		41/43/39/36	198 (49)	145 (49)	172 (77)	179 (55)	H:P—<0.001 ^a^ H:A2—0.027 ^a^ P:A1—0.008 ^a^ P:A2—0.005 ^a^
*High-density lipoprotein [mg/dl]*		41/43/39/36	53 (18)	53 (26)	49 (21)	48 (24)	H:A1—0.049 ^a^
*Triglycerides [mg/dl]*		41/43/39/36	136 (91)	89 (74)	136 (127)	120 (130)	P:A1—0.006 ^a^ P:A2—0.030 ^a^
*HbA1c [%]*		38/43/35/36	5.5 (0.5)	5.7 (0.6)	5.7 (0.5)	5.8 (0.8)	H:P—0.021 ^a^ H:A1—0.008 ^a^ H:A2—0.002 ^a^
*C-reactive protein [mg/dl]*		41/43/39/36	0.18 (0.28)	0.20 (0.39)	0.27 (0.45)	0.24 (0.43)	n.s. ^a^
*Maximal AAA diameter [mm]*		40/35			51.8 (10.9)	47.8 (12.9)	n.s. ^a^
*Aortic segment volume [cm³]*		28/31			121.6 (95.7)	93.3 (70.3)	n.s. ^a^
*Maximal ILT diameter [mm]*		30/31			15.1 (14.8)	16.6 (14.1)	n.s. ^a^
*ILT volume [cm³]*		27/31			42.5 (77.7)	35.5 (44.6)	n.s. ^a^
*Categorical variables*		n	n (%)	n (%)	n (%)	n (%)	
*Sex*	Men	41/43/41/37	32 (78.0)	34 (79.1)	32 (78.0)	31 (83.8)	n.s. ^b^
	Women		9 (22.0)	9 (20.9)	9 (22.0)	6 (16.2)	
*Smoking*	Never	41/43/41/36	13 (31.7)	3 (7.0)	3 (7.3)	2 (5.6)	H:P—0.016 ^b^ H:A1—0.019 ^b^ H:A2—0.020 ^b^
	Past		17 (41.5)	24 (55.8)	21 (51.2)	18 (50.0)	
	Current		11 (26.8)	16 (37.2)	17 (41.5)	16 (44.4)	
*Hypertension*		41/43/41/37	22 (53.7)	39 (90.7)	38 (92.7)	28 (75.7)	H:P—<0.001 ^b^ H:A1—<0.001 ^b^ H:A2—0.043 ^b^ A1:A2—0.038 ^b^
*Hyperlipidaemia*		40/43/41/37	10 (25.0)	31 (72.1)	34 (82.9)	32 (86.5)	H:P—<0.001 ^b^ H:A1—<0.001 ^b^ H:A2—<0.001 ^b^
*Peripheral artery disease*		41/43/41/37	0 (0.0)	43 (100.0)	0 (0.0)	13 (35.1)	H:P—<0.001 ^b^ H:A2—<0.001 ^b^ P:A1—<0.001 ^b^ P:A2—<0.001 ^b^ A1:A2—<0.001 ^b^
*Coronary heart disease*		41/43/41/37	0 (0.0)	19 (44.2)	21 (51.2)	18 (48.6)	H:P—<0.001 ^b^ H:A1—<0.001 ^b^ H:A2—<0.001 ^b^
*Myocardial infarction*		41/43/40/37	0 (0.0)	10 (23.3)	12 (30.0)	9 (24.3)	H:P—0.001 ^c^ H:A1—<0.001 ^b^ H:A2—0.001 ^c^
*Stroke*		41/43/41/37	0 (0.0)	6 (14.0)	4 (9.8)	3 (8.1)	H:P—0.026 ^c^
*Vascular Stent*		41/43/40/36	0 (0.0)	27 (62.8)	10 (25.0)	10 (27.8)	H:P—<0.001 ^b^ H:A1—0.002 ^b^ H:A2—0.001 ^b^ P:A1—0.001 ^b^ P:A2—0.005 ^b^
*Diabetes mellitus type 2*		41/43/41/37	5 (12.2)	11 (25.6)	12 (29.3)	12 (32.4)	H:A2—0.031 ^b^
*Chronic obstructive pulmonary disease*		41/43/41/37	1 (2.4)	14 (32.6)	9 (22.0)	11 (29.7)	H:P—<0.001 ^b^ H:A1—0.007 ^b^ H:A2—<0.001 ^b^
*Antiplatelet therapy*		41/43/41/37	6 (14.6)	38 (88.4)	38 (92.7)	33 (89.2)	H:P—<0.001 ^b^ H:A1—<0.001 ^b^ H:A2—<0.001 ^b^
*Anticoagulation therapy*		41/43/41/37	3 (7.3)	7 (16.3)	5 (12.2)	5 (13.5)	n.s. ^b^
*Antihypertensive therapy*		41/43/41/37	22 (53.7)	36 (83.7)	38 (92.7)	33 (89.2)	H:P—0.003 ^b^ H:A1—<0.001 ^b^ H:A2—0.001 ^b^
*Lipid-lowering agents*		41/43/41/37	7 (17.1)	39 (90.7)	35 (85.4)	34 (91.9)	H:P—<0.001 ^b^ H:A1—<0.001 ^b^ H:A2—<0.001 ^b^

Abbreviations: AAA, abdominal aortic aneurysm; ILT, intraluminal thrombus; IQR, interquartile range; PAD, peripheral artery disease; n, number of individuals or available data sets; n.s., not significant; H, healthy control cohort; P, PAD patient cohort; A1, AAA patient cohort 1; A2, AAA patient cohort 2. Values are given as medians with IQRs for continuous variables, or n (%) for categorical variables. The following statistical tests were applied: ^a^ Wilcoxon’s rank-sum test, ^b^ chi square test, ^c^ Fisher’s exact test.

**Table 2 jcm-12-07558-t002:** Explorative parameters of study groups.

*Parameter*		*Healthy* *(n = 41)*	*PAD* *(n = 43)*	*AAA Cohort 1* *(n = 41)*	*AAA Cohort 2* *(n = 37)*	*p-Value*
*Metric variables*	n	Median (IQR)	Median (IQR)	Median (IQR)	Median (IQR)	
*MPO [ng/mL]*	41/43/41/37	9.16 (6.57–13.63)	11.74 (8.70–13.64)	13.63 (10.92–22.19)	14.36 (10.08–19.73)	H:P—0.055 H:A1—<0.001 H:A2—<0.001 P:A1—0.004 P:A2—0.034 A1:A2—0.462
*D-dimer [µg/mL]*	41/43/41/37	0.38 (0.00–0.51)	0.58 (0.42–0.97)	1.27 (0.55–1.73)	0.90 (0.52–1.60)	H:P—<0.001 H:A1—<0.001 H:A2—<0.001 P:A1—0.006 P:A2—0.070 A1:A2—0.258
*Score*	41/43/41/37	−0.93 (−1.55–−0.09)	−0.19 (−0.73–0.71)	1.26 (0.12—4.06)	1.07 (−0.35—2.34)	H:P—0.002 H:A1—<0.001 H:A2—<0.001 P:A1—<0.001 P:A2—0.014 A1:A2—0.248

Abbreviations: AAA, abdominal aortic aneurysm; IQR, interquartile range specified as 25th–75th percentile; PAD, peripheral artery disease; n, number of individuals or available data sets; H, healthy control cohort; P, PAD patient cohort; A1, AAA patient cohort 1; A2, AAA patient cohort 2. Values are given as medians with IQRs, *p*-values refer to Wilcoxon’s rank-sum test.

**Table 3 jcm-12-07558-t003:** Multivariable analysis (binary logistic regression) of AAA diagnosis (AAA1) versus non-AAA (healthy + PAD) including major risk factors and comorbidities.

*Parameter*	*Exp(B)*	*95% CI Lower Value*	*95% CI Upper Value*	*p-Value*
*Score (dichotomised)*	22.296	5.841	85.113	<0.001
*Smoking (ever)*	7.941	1.115	56.559	0.039
*Hypertension*	2.670	0.465	15.344	0.271
*Hyperlipidaemia*	10.880	2.701	43.823	0.001
*Coronary heart disease*	13.196	2.201	79.126	0.005
*Myocardial infarction*	1.577	0.294	8.463	0.595
*Stroke*	0.535	0.080	3.551	0.517
*Vascular Stent*	0.041	0.007	0.241	<0.001
*Chronic obstructive* *pulmonary disease*	0.528	0.131	2.132	0.370
*Constant*	0.001			<0.001

Abbreviations: CI, confidence interval; Exp(B), odds ratio.

## Data Availability

The data that support the findings of this study are available from the corresponding author, C.B., upon reasonable request.

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
