# Peer review of "Diagnostic Utility of a Combined MPO/D-Dimer Score to Distinguish Abdominal Aortic Aneurysm from Peripheral Artery Disease"

_jcm, 2023, doi:10.3390/jcm12247558_

Round 1

Reviewer 1 Report

Comments and Suggestions for Authors

The specificity of an elevated MPO score to diagnose AAA is largely influenced by selection of subjects. Thus demographics and cardiovascular status of subjects in the different cohorts must be presented as detailed as possible.  

Comments on the Quality of English Language

Acceptable language.

Author Response

We would like to thank the reviewer and sincerely appreciate the comments and points of critique raised. The manuscript has been modified accordingly. The text with tracked changes is submitted and the point-by-point reply is given below.

Reviewer #1

Introduction: The introduction gives the reader a convincing background.

Method: The recruitment process and the imaging for settling diagnosis required further explanation.

P3, Line 90-91: ….confirmed hospital disease record were enrolled”. Did you only rely on a diagnostic code for peripheral vascular disease typed in the record? What was the minimal requirement to be selected to the PAD cohort or get the PAD diagnosis? Please state this clear in method paragraph! 

Reply: The requirement for inclusion was PAD diagnosis as attested in Austrian health care records and an available pelvis/leg CTA scan. Disease stage ranged from I to IV (20% stage I, 3% stage IIa, 60% stage IIb and 17% stage IV) which is now further specified in the Methods section, line 89.

P3 Line 91-95: Who were the recruited healthy controls from different hospital clinics? Staff / previous staff? Clinical patients?
Did you perform any objective examinations to confirm that all subjects in the “healthy cohort” were in acceptable cardiovascular health? In your previous paper (Thromb Haemost 2019)) several subjects with diabetes or history of cardiovascular events were enrolled in the reference group that was named “healthy”.

Reply: Healthy controls were primarily recruited from General Surgery, Urology and Ophthalmology patients (presenting for routine check-ups) with ultrasound-confirmed absence of AAA. As they were age-matched with the AAA collective, various chronic conditions frequent in advanced age (such as hypertension, hyperlipidaemia or diabetes) were not considered an indication for exclusion. Yet clinically manifest cardiovascular disease in terms of MI, stroke, stents or clinically documented PAD and CHD were applied as exclusion criterion. This has now been specified more clearly in the Methods section, line 91. Furthermore, routine blood parameters (assessed on the day of plasma preparation) have been added to Table 1 to facilitate comparison of overall health conditions and indicate e.g. that the healthy controls had significantly lower HbA1c blood levels than the patient cohorts.

P3 Line 96-97: “The aneurysm morphology of AAA patients was determined using computed tomography angiography (CTA) scans”. Did the CTA scan focus only on the thoracic and abdominal region or the whole lower limbs down to the pedal arteries? You divided AAA:s further into two subgroups depending on additional diagnosis. According to my experience most AAA subjects have atherosclerotic plaques along the lower extremity arteries. If the intension was to split AAA:s into two categorize with different levels of atherosclerotic burden, why just pay attention to flow restricted pathology in the lower extremities?  

Reply: CTA scans were generally focused on the thoracic and abdominal region (including upper extremity) and only occasionally extended to the lower limbs – as now more clearly stated in the Methods section, line 100. The design of the two AAA collectives was based on the following rationale: Cohort AAA1 did not include patients with diagnosed PAD to allow for a clearer separation between individuals with clinically manifest aneurysmal versus peripheral athero-occlusive disease in exploratory biomarker analysis. Exclusion criteria for cohort AAA2 were less stringent to reflect the usual frequency of concurrent AAA and PAD and to validate the diagnostic biomarkers under co-morb conditions. Yet, with respect to overall atherosclerotic burden, the AAA and PAD collectives did not differ in the frequency of CHD, MI or stroke (Table 1). This explanation has now been included in the Discussion section, line 232.

P3 Line 98-99: “The PAD cohort had a pelvis/leg CTA scan or comparable imaging in the recent past, which attested to them being negative for AAA”. This means that some subjects in the PAD cohort may have peripheral arterial aneurysms as long as their AA diameter was below 30 mm. Describe the severity of PAD better in your PAD cohort and state more clearly in the method section what is the border between PAD versus non-PAD in the continuum from normal arterial walls ending to multilevel arterial occlusive disease.

Reply: The distribution of PAD disease stages has now been included in the Methods section, line 90. Peripheral arterial aneurysms were not considered an exclusion criterion (as now stated in Study Limitations, line 312) but would be apparent for lower limbs in pelvis/leg CTA analysis. With respect to cerebral artery aneurysms, a reportedly low prevalence of up to 5% (2 persons per study group) could be expected (PMID: 16943405).

It seems that all patients had CTA / CT or comparable imaging while AA ultrasound scanning ruled out AAA in controls.
Q: Was all vascular examinations and image analysis performed after study enrollment according to research specific protocol, close in time to the venous blood sample? Alternatively: Were you compiling data from historical clinical CT reports found in the archive? Explain this in the method paragraph.

Reply: CTA measurements of AAA patients were generally conducted on the day of venous blood sampling. To exclude the presence of AAA in healthy controls and PAD patients, an ultrasound analysis was performed at the time of study inclusion, unless a recent (≤ 6 months) imaging record was available (as now specified in Methods, line 100).

Results:

P 5-6, Table 1
The table give the reader a nice overview but focus mainly on categorical data.
The cohorts with diagnosed vascular disease are well matched for most presented variables.
What basic clinical / demographic data are never presented in the table or elsewhere in the manuscript?
1) Obesity is a risk factor for cardiovascular disease, metabolic syndrome and is associated with low grade inflammation / activation of the immune system.

I suggest that you add at least BMI or waist circumference data in the table for subgroup comparison.
How is circulating plasma MPO and D-dimer related to body composition or BMI?

Reply: BMI values have now been added to Table 1 and differ moderately between PAD and AAA cohorts. Yet, there was no correlation between BMI and any of the explorative parameters as now mentioned in the Discussion section, line 290.

2) It is surprising that you abstain from presenting segmental blood pressure data. ABI data is reasonable to show when cohorts with vascular disease are involved, at least the number of subjects with pathological low / high ABI in each cohort should be presented. Successful vascular interventions can of course mask the degree of PAD. 

Are reported number of medical diagnosis and the drug consumption based on self-reporting or review of individual medical records?  

Reply: Unfortunately, ABI values were not systematically included in the data collection for the study groups – which we have now acknowledged under limitations (Discussion section, line 314). Moreover, as you kindly point out, successful PAD interventions may mask the degree of PAD in terms of ABI values.

Regarding medical diagnosis and drug consumption, both the structured questionnaire for patient self-reporting and the medical records were consistently evaluated, and discrepancies were solved by contacting patient or primary care physicians, as now specified in the Methods section, line 108.

You explain abbreviations well. Stroke and stents are placed closed together in the table. Do you count all kinds of stents within and without the vascular system?

Reply: Vascular stents were recorded – as now explained in Table 1.

How do you define hyperlipidemia? Review of the medical record show elevated plasma level in a laboratory test anytime in the past? Self-reported “YES” in the questionary?

Reply: Also in this respect, the structured questionnaire for patient self-reporting and the medical records were evaluated, and discrepancies were solved by contacting patient or primary care physician (Methods section, line 108).

Are subject with history of myocardial infarction also automatically added to the coronary heart disease group?

Reply: Yes – with the exception of 1 patient where CHD could not be ascertained as the cause of MI.

P7 Table 2

It is easy to misunderstand the IQR numbers presented within the parentheses because median ± half the IQR number give us the lower and upper limit of the boxplot. The same results are shown in figure 1 and table 2, easy to get confused. Have you thought about presenting Q1 and Q3 numbers instead of IQR in table2?

Reply: Thank you for the suggestion. We have now substituted the IQR by the values of 25th and 75th percentile in Table 2.

Discussion:

Metric data of aneurysm size and thrombus size are shown in table 1. Was concentration of blood parameters related to AAA size or AAA thrombus volume? It is reasonable to present some statistical analysis or explain why you abstain from discussing this matter.

Reply: Correlations of the three explorative parameters with aneurysm and thrombus size were included in our previous publication (PMID: 30822810): While D-dimer showed highest correlation with ILT dimensions, all three parameters – and in particular the newly developed score – were significantly associated with AAA diameter and growth. Since the comparison with PAD patients does not affect these associations within the AAA cohort, we did not include the analyses in the current manuscript but have now made a reference to our previous study in the Introduction, line 69.

P9 Line 234 “PAD is characterized by a stenotic process resulting from atherosclerotic occlusion of the lumen in a rigid blood vessel”. You claim that the arterial wall is rigid in PAD but mention nothing about the arterial wall stiffness in AAA.

Reply: We have largely changed the Discussion section and the respective sentence now reads as follows (line 244): “While the blood vessel wall is largely devoid of the elastic recoil properties in both disorders, AAA is mainly driven by an expansile and degenerative process [26] whereas PAD is characterized by a stenotic process resulting from atherosclerotic occlusion of the blood vessel lumen [27].”

Line 235-240 Discuss release of inflammatory mediators to the circulation in atherosclerotic versus aneurysmal disease. According to table 1 symptomatic coronary artery disease and cerebrovascular disease are equally frequent in patients regardless of cohort. Does it matter if the arterial occlusive disease is found in the limbs or at other sites for the degree of the inflammatory response?

Reply: Based on the largely comparable profile of co-morbidities between AAA and PAD patients, we would suggest that potential confounders might be less prominent in chronic than in acute cardiovascular disease settings (irrespective of the anatomic location), in particular with regard to the release of inflammatory mediators to circulation. To explain this notion, we included the following paragraph in the Discussion, line 268:

“Yet, acute events such as atherosclerotic plaque rupture associated with subsequent ischemia must be recognized as a potential confounding factor for the interpretation of D-dimer, MPO as well as the combined score. A prominent example would be an acute coronary event leading to myocardial infarction in which case the plasma concentration of MPO may be about ten times as high as the values measured in our study [15]. Similarly, circulating D-dimer is also found increased in myocardial infarction ranging at similar values as reported in our study [43,44]. Thus, any condition causing a major acute inflammatory response with neutrophil activation, or a thrombotic event associated with a spike in D-dimer needs to be taken into consideration when interpreting the MPO/D-dimer based score in relation to PAD and AAA.”

P9 Line 252- “In daily clinical practice the distinction of patients with AAA from atheroocclusive disease such as PAD may be relevant and a simple test raising concern for AAA could prove useful”. I agree that an accurate new simple pre-imaging test could be useful. Is the MPO score test accessible, specific and cheap enough as the first line aneurysmal screening test in an unselected large cohort with cardiovascular risk factors? Are your present results so promising that is it worth investigating it further? I suggest that you rewrite and create a revised short paragraph by edit present line 282-283. See next comment that is related.

P10 Line 282-283 “Lastly, increasing the number of study participants and evaluation of the studied variables at other institutions is required to confirm the usefulness of the proposed score in clinical practice”. See also the previous comment. This sentence can be rewritten and extended to make more sense. The performance of the lab parameters and their combined score also needs to be evaluated against other circulating biomarkers and physical examination in a blinded manner before ultrasound screening give us the key.

Reply: We have modified the Discussion text (line 321) to clarify the potential usefulness of the score as a diagnostic signal to identify AAA patients among those with PAD.

“In daily clinical practice with increasing workloads and complex diagnostic situations, diagnostic scores may provide useful signals to alert clinicians to a potentially high-risk condition such as AAA and trigger specialist imaging. Our data suggest that it may be possible to use simple parameters to identify patients with AAA even among individuals with advanced vascular comorbidity such as PAD. While D-dimer is a routine parameter ordered frequently in patients with suspected or known cardiovascular condition, plasma MPO is less commonly determined but widely available in clinical laboratories. The combined expenses for these diagnostic tests is within the cost range of abdominal ultrasound, and considerably lower than for computed tomography. However, the performance of these blood parameters and their combined score needs to be further evaluated in larger multicentre studies, against other circulating biomarkers and physical examination in a blinded manner, to confirm the usefulness of the proposed score as a simple pre-imaging test.”

By the changes introduced we hope to have adequately addressed the reviewers´ concerns and substantially improved the quality of our manuscript. We therefore hope to meet the quality standards for publication in the Journal of Clinical Medicine and we look forward to hearing from you.

Yours sincerely, Christine Brostjan

Reviewer 2 Report

Comments and Suggestions for Authors

Diagnostic utility of a combined MPO/D-dimer score to discern abdominal aortic aneurysm from peripheral artery disease

Methods

1.       Pg 3. Lines 96-102. Does the presence of D-dimer/MPO without clinical evidence of AAA mean that these patients will go on to develop AAA?

Results

1.       Pg 5. Lines 134, 143. What does Error! Reference source not found mean or refer too?

2.       Pg 6. Line 171-3. Given these results it is hard to justify these markers being reliable for discerning between AAA and PAD. However, when the combined score is used AAA2 vs PAD has a significant difference. Can you make sure this is thoroughly explained in the Discussion?

3.       There were only thirteen patients in the AAA2 group that also had PAD is this correct? It is questionable if this is enough to generate a reliable power for statistical significance.

Discussion

1.       Pg 9. Lines 228-243. You need to be more specific as to what the cellular and molecular differences that are known are between PAD and AAA. Everyone knows the difference between and aneurysm and occlusive disease. What are the known blood character differences between these two?

Author Response

We would like to thank the reviewer and sincerely appreciate the comments and points of critique raised. The manuscript has been modified accordingly. The text with tracked changes is submitted and the point-by-point reply is given below.

Reviewer #2

Methods

  1. Pg 3. Lines 96-102. Does the presence of D-dimer/MPO without clinical evidence of AAA mean that these patients will go on to develop AAA?

Reply: Unfortunately, we don´t have any indication or data to support this interesting notion, i.e. to answer this question a study design distinct from ours would be required.

Results

  1. Pg 5. Lines 134, 143. What does Error! Reference source not found mean or refer too?

Reply: Links to the respective tables and figures had been included, which were apparently lost (converted to error messages) in the MDPI formatted version. We apologize and have removed the links, i.e. corrected the errors.

  1. Pg 6. Line 171-3. Given these results it is hard to justify these markers being reliable for discerning between AAA and PAD. However, when the combined score is used AAA2 vs PAD has a significant difference. Can you make sure this is thoroughly explained in the Discussion?

Reply: Thank you for alerting us to this shortcoming. We have now tried to explain the findings (differences between separate parameter analyses versus combined score) more clearly in the Discussion section, line 283.

  1. There were only thirteen patients in the AAA2 group that also had PAD is this correct? It is questionable if this is enough to generate a reliable power for statistical significance.

Reply: The rationale for the two AAA collectives is now more clearly stated in line 232. Thus, we would like to emphasize that cohort AAA2 was intended as a second (validation) group (reflecting the usual frequency of concurrent AAA and PAD) rather than being powered to distinguish between AAA patients with or without concomitant PAD.

Discussion

  1. Pg 9. Lines 228-243. You need to be more specific as to what the cellular and molecular differences that are known are between PAD and AAA. Everyone knows the difference between and aneurysm and occlusive disease. What are the known blood character differences between these two?

Reply: The clinical risk factors (smoking, hypertension and hyperlipidaemia) as well as the profile of circulating cellular and humoral factors in AAA and PAD overlap to a large extent. This may be due to the chronic inflammatory nature of both conditions. We have expanded this section as follows

Line 244: Yet, the two disease conditions also differ in various aspects of their pathogenesis. While the blood vessel wall is largely devoid of the elastic recoil properties in both disorders, AAA is mainly driven by an expansile and degenerative process [26] whereas PAD is characterized by a stenotic process resulting from atherosclerotic oc-clusion of the blood vessel lumen [27]. Thus, the lipidomic profile was suggested to differ between AAA and PAD [39]. With respect to vessel wall destruction, extracellular matrix degrading factors such as matrix metalloproteinases (MMP)-2 and 9 and their inhibitors may therefore be more prominent in AAA than PAD. Yet, the potential of serum MMP levels as biomarkers seems limited for either condition [40,41]…

Line 257: Consequently, the profile of circulating inflammatory factors overlaps in part between AAA and PAD, i.e. C-reactive protein, interleukin (IL)-1 and IL-6 were consistently found associated with both conditions [34-36]. Yet, serum complement factor C5a, which functions as a chemotactic signal for neutrophils, is reportedly higher in AAA compared to PAD patients [37,38]. Also various other neutrophil-derived mediators have been investigated as potential diagnostic biomarkers. However, direct comparisons between AAA and athero-occlusive disease are largely missing.

By the changes introduced we hope to have adequately addressed the reviewer´s concerns and substantially improved the quality of our manuscript. We therefore hope to meet the quality standards for publication in the Journal of Clinical Medicine and we look forward to hearing from you.

Yours sincerely, Christine Brostjan

Reviewer 3 Report

Comments and Suggestions for Authors

The topic is of interest. However, the main drawback is the potential novelty of the data presented. In this respect, several papers including a paper of the same group have addressed the role of D-dimer and MPO as diagnostic and/or prognostic biomarker of AAA.

1.- The authors have recently published an article addressing the role of the same score in AAA patients vs healthy subjects (Thromb Haemost 2019 May;119(5):807-820). As the number of subjects is similar to the AAA1 cohort present here (n=41 for H and n=41 for AAA), it should be important to clarify this issue. If they are the same analysis, I would suggest to focus on the AAA2 cohort if this is completely new data. In this respect, the comparison of AAA2 cohort vs healthy should be tested to see if the previous data is validated. Moreover, the potential association of the score with AAA progression should be also tested in AAA2 cohort if possible.

2.- Why exclusion criteria for AAA2 was different?

3.- The comparison of the diagnostic performance by AUC should be done by c-statistics.

4.-  Missing reference: Plasma D-dimer as a predictor of intraluminal thrombus burden and progression of abdominal aortic aneurysm Life Sci. 2020 Jan 1:240:117069. doi: 10.1016/j.lfs.2019.117069. Epub 2019 Nov 18.

Author Response

We would like to thank the reviewer and sincerely appreciate the comments and points of critique raised. The manuscript has been modified accordingly. The text with tracked changes is submitted and the point-by-point reply is given below.

Reviewer #3

The topic is of interest. However, the main drawback is the potential novelty of the data presented. In this respect, several papers including a paper of the same group have addressed the role of D-dimer and MPO as diagnostic and/or prognostic biomarker of AAA.

1.- The authors have recently published an article addressing the role of the same score in AAA patients vs healthy subjects (Thromb Haemost 2019 May;119(5):807-820). As the number of subjects is similar to the AAA1 cohort present here (n=41 for H and n=41 for AAA), it should be important to clarify this issue. If they are the same analysis, I would suggest to focus on the AAA2 cohort if this is completely new data. In this respect, the comparison of AAA2 cohort vs healthy should be tested to see if the previous data is validated. Moreover, the potential association of the score with AAA progression should be also tested in AAA2 cohort if possible.

Reply: No, the analyses are distinct from our previous study (PMID: 30822810), since a one-to-one matching was performed among the 4 groups based on age (±2 years) and sex. This is now also indicated in the Methods section, line 110.

2.- Why exclusion criteria for AAA2 was different?

Reply: The design of the two AAA collectives was based on the following rationale (as now explained in the Discussion, line 232): Cohort AAA1 did not include patients with PAD to allow for a clear separation between individuals with clinically manifest aneurysmal versus (peripheral) athero-occlusive disease in exploratory biomarker analysis. Exclusion criteria for cohort AAA2 were less stringent to reflect the usual frequency of concurrent AAA and PAD and to validate the diagnostic biomarkers under co-morb conditions.

3.- The comparison of the diagnostic performance by AUC should be done by c-statistics.

Reply: We have performed the suggested comparison of AUC curves (referring to Figure 2) and found a significant difference between score and MPO which is now mentioned in the Results section, line 216.

4.-  Missing reference: Plasma D-dimer as a predictor of intraluminal thrombus burden and progression of abdominal aortic aneurysm Life Sci. 2020 Jan 1:240:117069. doi: 10.1016/j.lfs.2019.117069. Epub 2019 Nov 18.

Reply: We apologize for the shortcoming and have now included the reference (no. 10).

By the changes introduced we hope to have adequately addressed the reviewer´s concerns and substantially improved the quality of our manuscript. We therefore hope to meet the quality standards for publication in the Journal of Clinical Medicine and we look forward to hearing from you.

Yours sincerely, Christine Brostjan

Round 2

Reviewer 2 Report

Comments and Suggestions for Authors

well written 

Reviewer 3 Report

Comments and Suggestions for Authors

None